# Diversity, Abundance and *Leishmania infantum* Infection Rate of Phlebotomine Sandflies in an Area with Low Incidence of Visceral Leishmaniasis in Northern Tunisia

**DOI:** 10.3390/microorganisms10051012

**Published:** 2022-05-11

**Authors:** Marwa Weslati, Jamila Ghrab, Meriem Benabid, Olfa Souissi, Karim Aoun, Aïda Bouratbine

**Affiliations:** 1«Laboratoire de Recherche Parasitoses Médicales, Biotechnologies et Biomolécules, LR 16IPT06 and LR 20IPT06», Institut Pasteur de Tunis, Université Tunis El-Manar, Tunis 1002, Tunisia; marwa.weslati@pasteur.utm.tn (M.W.); abdelkafi.jamila@gmail.com (J.G.); meriembenabid@gmail.com (M.B.); souissiolfa75@yahoo.fr (O.S.); karim.aoun@pasteur.rns.tn (K.A.); 2Institut Supérieur des Sciences et Technologies de l’Environnement de Borj Cedria, Université de Carthage, Carthage 1054, Tunisia

**Keywords:** *Leishmania infantum*, *Phlebotomus perniciosus*, *Phlebotomus perfiliewi*, diversity, visceral leishmaniasis, quantitative PCR, molecular typing

## Abstract

We report the study of sandfly *Leishmania* infection in an area of low incidence of visceral leishmaniasis in Tunisia. Sandflies were collected monthly using CDC light-traps set in houses and animal shelters during May–November 2016 and 2017. All males were identified at the species level. A sample of 878 females including all gravid specimens was subjected to kDNA qPCR for *Leishmania* detection and parasite load estimation. *Leishmania* species were determined by ITS1 PCR sequencing, and species identification of infected sandflies was performed by DNA barcoding. *Phlebotomus perfiliewi* and *P. perniciosus* were the dominant species during the two-year period. However, comparison of their relative abundances showed that *P. perniciosus* was more abundant during peaks of 2017 with longer activity duration. Real-time kDNA PCR did not detect *Leishmania* infection in 2016, although it identified four positive specimens (0.7%) in 2017. All four infected specimens were identified as *P. perniciosus*. ITS1 PCR sequencing allowed *L. infantum* identification in one kDNA qPCR-positive specimen. This was a *P. perniciosus* gravid female with a high parasite load caught during the long-lasting peak of 2017. This work highlights the usefulness of multi-seasonal studies of sandfly dynamics and kDNA qPCR in screening *Leishmania* infection and determining *L. infantum* vectors in hypo-endemic foci of human leishmaniasis.

## 1. Introduction

Throughout the Mediterranean basin, *Leishmania* (*L.*) *infantum* is the causative agent of visceral leishmaniasis (VL) and also causes cases of cutaneous leishmaniasis [1]. Domestic dogs are the main reservoir hosts and a dozen of *Phlebotomus* species are involved in transmission [2,3,4]. Among potential vector species, three, namely, *Phlebotomus ariasi*, *P. perniciosus*, and *P. perfiliewi*, all members of the *Larroussius* subgenus, have been incriminated as vectors according to conventional criteria in the western Mediterranean region [5,6,7]. In this Mediterranean region, seasonal dynamics of competent vectors showed a regular bell-shaped density curve with a broad central peak encompassing the period July–September and falling between the beginning of May and the end of October [8].

In Tunisia, which is localized in the Southern part of western Mediterranean basin, *L. infantum* is known to be involved in canine leishmaniasis and both cutaneous and VL in humans [9]. The cutaneous form prevails in the Northern part of the country while VL is distributed in the North and the Center [10,11]. In this large VL endemic area, the spatial study of disease incidence by district identified a hotspot region where VL was homogeneously endemic at a medium/high magnitude. This VL hotspot encompassed the known historic northern VL foci located in the semi-arid zone and supported the presence of a spatial environmental factor underlying the level of endemicity in the area [12]. The entomological survey showed that *P. perniciosus* was the almost exclusive *Larroussius* species in the hot spot foci [13]. It was very abundant, especially inside houses, with long seasonal activity and a high *L. infantum* infection rate [13]. On the other hand, the same spatial study showed that VL incidence rates in districts outside the hotspot were highly variable and not spatially related to each other; the high and low incidence rates in these districts were most likely the result of local factors [12]. This was the case of the emerging VL foci located in the arid bioclimatic zone of Central Tunisia that evidenced a high mobilization of water resources recent years [12,14]. In these emerging foci of high VL incidence, several entomological surveys have been carried out showing *L. infantum* infection of several *Laroussius* species, notably *P. perniciosus*, *P. perfiliewi*, and *P. longicuspis* [15,16,17], but also showed *L. infantum* infection of *Phlebotomus papatasi* [15,17] and *Sergentomyia minuta* [15]. These studies have also highlighted the role of *P. perfiliewi* in these particular foci [15,16]. To our knowledge, no information is available on *L. infantum* infection rate of phlebotomine sandflies in Northern VL endemic areas of low incidence where *P. perniciosus* and *P. perfiliewi* are known to be sympatric species [18].

Quantitative polymerase chain reaction (qPCR) targeting minicircle kinetoplast DNA (kDNA) represents a suitable tool for the diagnosis of *Leishmania* in vectors [19]. In fact, kDNA qPCR is a sensitive method that allows a detection limit of one flagellate per dissected gut, which legitimates its use as a screening method [20]. Furthermore, kDNA qPCR can determine parasite load [20], with a high burden being correlated with strong evidence of *Leishmania* transmission [21]. On the other hand, most sandfly species require a blood meal to develop their eggs, which facilitates contact with infected and uninfected hosts and transmission of *Leishmania* [22]. Targeting gravid female specimens for infection study is therefore a good method to ensure the presence of sandflies with at least one ingested blood meal and increase the chance to find *Leishmania* [23].

The objective of this work was to study the diversity, abundance, and *L. infantum* infection rate of phlebotomine sandflies in an area with low incidence of visceral leishmaniasis in northern Tunisia and to discuss the usefulness of quantitative kDNA real-time PCR and molecular investigation of gravid female specimens in screening *Leishmania* infection and determining *L. infantum* vectors.

## 2. Materials and Methods

### 2.1. Study Area

The study was conducted in the governorate of Siliana, North-West Tunisia, and specifically in the district of Siliana-North (Figure 1). Average of VL incidence rates observed in this district during 1996–2006 was one of the lowest of the VL endemic area [12]. Furthermore, no VL cases originating from this district have been reported by the Regional Health Directorate during the period 2008–2017, with VL being a notifiable disease in Tunisia.

Weather conditions of the region are known to be those of a Mediterranean semi-arid bioclimate, marked by a high degree of continentality and characterized by typical vegetation series of *Pinus halepensis* and *Tetraclinus articulata*. However, climatic parameters are subjected to annual variations, affecting annual degree of aridity and arid season duration.

Monthly rainfall and mean temperature recorded by the National Meteorological Office from January 2016 to December 2017 at the meteorological station of Ramlia located in Siliana-North district (Figure 1) were used to establish the ombro-thermic diagram (Figure 1) and allowed for the computation of the annual aridity index (*Ia*) by the following formula:(1)$$Ia=\frac{Pa}{T+10\;}$$
where *Pa* (mm) corresponds to the annual precipitation and *T* (°C) corresponds to the annual average of temperature. The higher the value of this index, the lower is the aridity of the period. These parameters indicated that the climate in the study region was globally more arid in 2017 than in 2016. In fact, the annual aridity index dropped from 15.4 (in 2016) to 8.6 (in 2017), and the arid season was longer in 2017 than in 2016 (Figure 1).

The sampling station (36°10′ N/9°28′ E at 708 m above sea level) was located at Djema, a rural locality surrounded by a large cereal field and characterized by the presence of a variety of domestic animals (goats, sheep, chickens, dogs, donkeys) living in semi-open shelters or free in yards next to houses (Figure 2). The study was conducted on private land. The land’s owner gave permission to conduct the study on the site.

### 2.2. Sandfly Collection, Habitat, and Physiological State of Female Specimens

Sandflies were collected using CDC miniature light traps equipped with a fine net cage (John W. Hock Company, Florida, USA) during two successive entomological seasons (June–November 2016 and May–November 2017). CDC traps were set monthly during two consecutive nights (with replacement of traps after every night) in 3 habitats: one CDC indoor (ID), one outdoor (OD), and two in animal shelters (AS). They were set operating 1–2 h before sunset until 1–2 h after sunrise.

All collected sandfly specimens were stored in ethanol 70% then rapidly sorted by sex. All male specimens were conserved by habitat and date of capture for species identification. To identify physiological state of females, they were checked under stereomicroscope. Females were categorized according to their abdomen status and on the basis of blood digestion described by Dolmatova and Demina [22] into (i) unfed (non-distended abdomen without visible blood); (ii) fed (presence of blood meal in the abdomen); (iii) gravid (with abdomen full of eggs). For fed females, blood meal was classified on fresh (red fresh blood filling the abdomen) and digested (dark red to brown blood clot in the proximal part of the abdomen). All female specimens were stored individually for study of *Leishmania* infection.

### 2.3. Sandfly Species Identification and Study of Leishmania Infection

Sandfly male specimens were morphologically identified. Briefly, head and genitalia were removed and mounted in Marc-André solution (40 g of chloral hydrate, 30 mL glacial acetic acid, and 30 mL of distelled water) observed under a microscope and identified at the species level according to morphological characteristics of aedeagus, stylopodite, and coxopodite, as previously described [24].

DNA was extracted separately from each female individual. Briefly, the whole body of female specimens were washed twice with distilled water, and then DNA was extracted using the method described by Ready et al. (1991) [25] with minor modifications. DNA was eluted in 20 µL TE and stored at −20 °C until use.

#### 2.3.1. Molecular Screening of *Leishmania* Infection by kDNA qPCR

*Leishmania* infection was first screened by pools of 5 sandflies’ DNA. Then, DNAs from sandflies grouped into each positive pool were analyzed individually to determine the positive specimens in each pool and to quantify the parasite load of each specimen. kDNA qPCR was performed as described by Mary et al. [26]. Briefly, qPCR was conducted in a final volume of 25 µL by using a TaqMan universal master mixture (Applied Biosystems, Foster City, CA, USA) containing 100 mM of each primer (5′-CTTTTCTGGTCCTCCGGGTAGG-3′) and (5′-CCACCCGGCCCTATTTTACACCAA-3′), 50 mM of probe (FAM 5′-TTTTCGCAGAACGCCCCTACCCGC-3′), and 1 µL of DNA extract. The DNA was amplified in an Applied Biosystems^®^ (Applied Biosystems, Foster City, CA, USA) for 40 cycles at 95 and 60 °C. A dilution series of *Leishmania* DNA extracted from 5 × 10^6^  *L. infantum* promastigotes was used (from 50 to 0.005 parasites/µL) to generate a standard curve for *Leishmania* quantification. Real-time PCR was considered positive for a qPCR threshold corresponding to 0.01 parasites/PCR reaction for the pools and 0.05 parasites/PCR reaction for individual specimens. Taking into account the (i) pooling dilution, (ii) the amount of biological sample (1 μL of sample DNA), and (iii) the elution volume of the extracted DNA (20 μL), the selected qPCR threshold corresponded to one parasite DNA per sandfly specimen.

#### 2.3.2. Identification of *Leishmania* Species by ITS1 PCR Sequencing

All positive kDNA qPCR specimens were systematically amplified by ITS1 PCR as previously described [27] using LITSR (5′-CTGGATCATTTTCCGATG-3′) and L5.8S (5′-TGATACCACTTATCGCACTTA-3′) primers. Analysis on a 2% agarose gel was used to verify the amplified product size. ITS1 PCR products were purified by using ExoSAP (ThermoScientific, EU), and *Leishmania* species identification was performed by DNA sequencing using an ABI Prism^®^ Big Dye™ Terminator, Cycle Sequencing Ready Reaction Kit, and AB1 3130 sequencing system (ABI, PE Applied Biosystems), with the same primers used for PCR. The sequence (LiDj033F4) was deposited in GenBank under the accession number OL804107.

#### 2.3.3. Identification of Sandfly Species by DNA Barcoding

All positive kDNA qPCR female specimens as well as *P. perniciosus* and *P. perfiliewi* specimens collected and microscopically identified during our study were molecularly typed on the basis of DNA barcoding. The mitochondrial cytochrome c oxidase gene subunit 1 (COI) was amplified using primers LCO 1490 (5′ GGTCAACAAATCATAAAGATATTGG-3′) and HCO 2198 (5′TAAACTTCAGGGTGACCAAAAAATCA-3′) as previously described [28]. Analysis on a 1% agarose gel was used to verify the amplified product size (about 650 bp). PCR products were purified and cycle-sequenced as described previously. The generated sequences, Dj030F1, Dj032B1, Dj033F4, DJF01, DjM02, Dj033E10 and DjM01 were deposited in GenBank under the accession numbers OL814952, OL814953, OL814954, OL814955, OL819881, OL821654 and OL821655 respectively. MEGA software version 7.0.26 (Kumar, Stecher, and Tamura 2016) was used to conduct multiple sequence alignments and phylogenic analysis.

### 2.4. Data Analysis

#### 2.4.1. Species Diversity

To compare either species diversity in a given habitat over time or species diversity in the different habitats, two diversity indexes widely used in ecology, and their corresponding equitability indexes were used [29,30]. Shannon’s index (H’) takes into account rare species and was used to study the temporal changes in a giving habitat. The Simpson’s index (D) gives more weight to dominant species than to rare ones, and it was used to make a comparison between habitats.

#### 2.4.2. Statistical Analysis

Statistical analysis was performed using the MedCalc Statistical software (version 11.4.4.0). We compared median sandfly densities using the Mann–Whitney nonparametric test. The Spearman’s non-parametric correlation test was applied to establish the relationship between *P. perniciosus* and *P. perfiliewi* densities. The *chi*-squared test was used for comparison of relative abundances. Fisher’s exact test was applied to a 2 × 2 contingency table as an alternative to the *chi*-squared test to compare the proportion of qPCR-positive female sandflies in (i) gravid vs. fed specimens or (ii) gravid vs. unfed specimens. The significance level was set at 5% for all tests.

## 3. Results

### 3.1. Sandfly Fauna in the Study Site

During the two-year period, a total of 6125 specimens were collected: 2106 were collected during 2016 and 4019 during 2017; 5674 specimens were caught in AS, 243 in ID, and 208 in OD (Table 1). Cumulatively, 3336 females and 2789 males (sex-ratio = 0.84) were caught. Females outnumbered males during the two years of capture (Table 1).

Among the collected females, 2575 (77.2%) were unfed, 735 (22%) were fed, and only 26 (0.8%) were gravid. Physiological state of females and degree of blood digestion is reported in Table 2.

### 3.2. Species Diversity

On the basis of morphological identification of male specimens, six species were identified, namely, *P.* (*Larroussius*) *perniciosus* (n = 1355, 48.58%), *P.* (*Larroussius*) *perfiliewi* (n = 1281, 45.93%), *Sergentomyia* (*Sergentomyia*) *antennata* (n = 93, 3.33%), *P.* (*Paraphlebotomus*) *sergenti* (n = 48, 1.72%), *P.* (*Phlebotomus*) *papatasi* (n = 9, 0.32%), and *P.* (*Larroussius*) *longicuspis* (n = 3, 0.11%); *P. perniciosus* and *P. perfiliewi* corresponded to 94.5% of sandfly male specimens, whereas *P. longicuspis*, *P. papatasi*, *P. sergenti*, and *S. antennata* were rare species.

In AS, diversity was fairly steady during the two years (H’/H’max = 0.55 in 2016 and H’/H’max = 0.48 in 2017) with the dominance of the two species *P. perfiliewi* and *P. perniciosus* during the two-year period (D/Dmax = 0.52 in 2016 and D/Dmax = 0.59 in 2017) (Table 3). Indoor and OD ecological diversity was higher in 2016 than in 2017 (H’/H’max = 0.96 and H’/H’max = 1 in 2016 versus H’/H’max = 0.28 and H’/H’max = 0.21 in 2017). In 2017, *P. perniciosus* was the only dominant species in ID (pi = 0.9, D/Dmax = 0.23), whereas *P. perfiliewi* was the only dominant species in OD (pi = 0.95, D/Dmax = 0.15) (Table 3).

### 3.3. Sandfly Density

Sandfly density was computed, taking into account the number of CDCs set and number of nights of capture, and density was expressed as number of sandfly specimens per CDC.night. Globally, in the whole study site, sandfly density varied from 2 to 250 sandflies/CDC.night in 2016 and from 2 to 286 sandflies/CDC.night in 2017. Although the number of sandfly specimens collected in 2017 was higher than in 2016, no statistical differences were shown between median densities observed in 2016 and 2017.

The recorded densities in the different habitats during each year of collection are reported in Figure 3. In AS, the highest densities were recorded during July 2016 (487 sandflies/CDC.night) and August–September 2017 (535 and 356 sandflies/CDC.night, respectively). In indoor and OD, sandfly densities were usually low (≤56 sandflies/CDC.night). The highest density in OD was observed in August 2017, while the highest density in ID was shown in September 2017 (56 and 46 sandflies/CDC.night, respectively) (Figure 3).

### 3.4. Seasonal Dynamics

Seasonal density pattern of sandflies during the two-year period is reported in Figure 4. Globally, sandfly activity started in May–June and ended in October. However, in 2017, adults were caught during a longer period than in 2016. On the other hand, in 2017, a sharp bi-modal trend, with a first minor peak in June and a major peak in August–September, was clearly shown. Furthermore, during this second year of collection, the male/female ratio was higher during the early period of collection, whereas females outnumbered males at the end of the season (Figure 4).

Monthly *P. perniciosus* and *P. perfiliewi* male densities showed a significant positive correlation during the two-year period (Spearman’s coefficient ρ = 0.83, *p* = 0.001) and fairly the same seasonal density pattern with one major peak in July 2016 and two peaks during 2017 (a minor one in June and a major one in August–September) was shown (Figure 4). Nevertheless, some differences were observed between the two species. In fact, while the density of *P. perfiliewi* and its dynamics over time showed little variability during the two years of collection (density around 70 sandflies/CDC.night during a sharp peak in July 2016 or August 2017), the density of *P. perniciosus* showed more variability. In fact, the density of *P. perniciosus*, which was much lower than that of P. perfiewi during the 2016 peak (15 sandflies/CDC.night), rose by about five times (reaching around 70 sandflies/CDC.night), and its duration lengthened, resulting in an extended peak encompassing August–September 2017 (Figure 4). Furthermore, study of sandfly species abundance during peaks showed that relative abundance of *P. perfiliewi* during July 2016 was significantly higher than that of *P. perniciosus* (76.6% versus 13.6%) (*p* < 0.0001), whereas relative abundances of *P. perniciosus* during June, August, and particularly September 2017 were significantly higher than that of *P. perfiliewi* (58%, 58%, and 77.7% versus 24.7%, 41.8%, and 15.8%, respectively) (*p* < 0.0001).

### 3.5. Leishmania Infection

Among the 3336 collected females, 878 specimens (26.3%) were selected for DNA extraction (Table 4). Selection was conducted according to: (i) habitat (ID, OD, AS): almost all female specimens from ID and OD were used for DNA extraction, whereas about 20% of specimens from AS (where female number was much higher than ID and OD) were randomly selected; (ii) month and year of capture: almost all female specimens caught in June, August, September, and October 2016 and May, June, July, and October 2017 were used for DNA extraction, whereas about 20% of specimens collected either during the peak of July 2016 or the peak of August–September 2017 were randomly selected; and (iii) physiological state of females (unfed, fed, and gravid): all gravid females were selected for DNA extraction, whereas a comparable number of fed vs. unfed specimens were used to study infection. For fed females, degree of blood meal digestion was noted (Table 4).

Four specimens were positive by kDNA qPCR; no infection was found among the 311 specimens collected in 2016. The four kDNA qPCR-positive specimens were found among the 567 specimens (0.7%) collected in 2017. All four positive females were caught in September, one from ID and three from AS. Two females were fed, and two were gravid, and the two positive fed females harbored fresh blood (Table 4); infection rate was significantly higher in gravid (8.7%) than either in fed (0.8%) (*p* < 0.05) or in unfed (0%) (*p* < 0.01) specimens.

All infected sandfly specimens were identified as *P. perniciosus* on the basis of the 613 base pairs of mitochondrial cytochrome c oxidase (COI) gene subunit 1. In fact, the four sequences from infected sandflies collected during this study, Dj033E10, Dj033F4, Dj032B1 and Dj030F1 (gene bank accession numbers: OL821654, OL814954, OL814953, OL814952 respectively) showed homology higher than 98% with (i) the Mediterranean haplotypes of *P. perniciosus* available in GenBank and (ii) nucleotide sequences from *P. perniciosus* male and female specimens collected and microscopically identified during our study (DjM01 and DjF01, gene bank accession numbers: OL821655, and OL814955).

### 3.6. Leishmania Infantum Infection and Parasitic Load

Of the four kDNA qPCR-positive *P. perniciosus* females, only one was positive by ITS1 PCR; DNA sequencing identifying *L. infantum* (gene bank accession number: OL804107) (Table 5). This was a gravid specimen caught in AS during September 2017. It was highly infected with thousands of parasites (Table 5). The other three specimens were negative by ITS1 PCR, and *Leishmania* species identification was not possible using this target. All harbored less than 20 parasites (Table 5).

## 4. Discussion

Sandflies were collected monthly through two entomological seasons by CDC light traps. This sampling method commonly used in VL eco-epidemiological studies allows for the capturing of several species of the *Larroussius* subgenus that are the potential vectors of *L. infantum*, the causative agent of VL [18]. Moreover, light traps are useful to collect large numbers of samples, allowing for the measurement of the relative changes in abundance over time [8,31]. They are also useful for determining the temporal activity of species as they capture active sandflies [8,31]. This method, however, may be biased towards capture of unfed female specimens rather than fed and gravid ones. In fact, unfed females are more active in looking for a host to take blood meal, so they are more attracted by CDC traps [32].

Contrary to other studies, females outnumbered males during the two seasons of capture [8]. Furthermore, unfed specimens were the most abundant, whereas gravid females were rarely caught. This seems to be related to method of capture. However, it is important to note that female sandflies were not dissected and visually classified as unfed, fed, and gravid. Thus, a small volume of digested blood, as well as eggs in the first stages of development, may not be revealed and, therefore, numbers of fed and gravid females may be underestimated.

Species identification of male specimens showed that *P. perniciosus* and *P. perfiliewi* were the most abundant species in the study site localized in the semi-arid bioclimatic zone. It is well known that these two *Larroussius* species prevail within a specific bioclimatic continuum from the humid to arid zones of Western Mediterranean countries with inter-specific relationships, with *P. perfiliewi* and *P. perniciosus* preferences being for humid and semi-arid zones, respectively [18,33]. In Iran, *P. perfiliewi* is considered an intermediate species between xerophylious and hygrophylious species that prefer a humid region [34]. In Tunisia, global aridity of the center and the south of the country seems to limit the distribution of this species to the north above the “dorsal” upland [24]. However, *P. perfiliewi* was reported with high prevalence below the “dorsal” in irrigated zones [35]. This highlights the importance of micro-climate impact on sand fly fauna structure. On the other hand, the sampling station was located in anthropic rural biotope characterized by the presence of many domestic animals such as dogs, sheep, and chicken, which may explain the abundance of *Larroussius* species in the sample collection [18,31,36,37,38].

To study species diversity in the different habitats (AS, ID, and OD), the Shannon’s and Simpson’s diversity indexes and their correspondent equitability indices were computed. By using these indices, it was possible to follow the variation of the species diversity over time, but also to make comparison between habitats [39]. In AS, equitability indices showed that *P. perniciosus* and *P. perfiliewi* were dominant through the two-year period. Furthermore, the high sandfly density observed in AS likely determined the global trend in the study site [13,18]. Indeed, AS seems to constitute ecological niche of these sandfly species [40]. It is usually characterized by biotic and abiotic factors suitable for larval development; adult activity; and female’s rest such as organic matter produced by domestic animals, vertebrate host source of blood meals, suitable brightness, and humidity conditions. Being close to houses, this prolific habitat may maintain and control *Phlebotomine* abundance ID and OD, where vector contact with humans and dogs is the most frequent. In these two latter habitats, high species diversity was observed in 2016, and a switch toward a quasi-exclusive presence of *P. perfiliewi* OD and *P. perniciosus* ID was shown in 2017.

Globally, adult sandfly activity in the studied site started in May–June and extended to October with a large variation in sandfly densities during the season, which was in accordance with phenology studies performed in temperate Mediterranean region [8]. The bimodal trend observed in study site in 2017 likely indicated the occurrence of two sandfly generations at least and was already reported for both *P.*
*perniciosus* and *P. perfiliewi* in Tunisia [13,41]. Interestingly, the multi-seasonal entomological survey evidenced two different monthly sandfly density patterns according to the year of collection. In 2016, a sharp peak of abundance was observed in July, while a long-lasting peak was observed during August–September 2017. Moreover, the magnitude of density peak observed in 2016 was mainly related to *P. perfiliewi*, whereas the long duration of the major peak observed in 2017 was associated with *P. perniciosus.* These specific species sandfly phenologies according to the year of capture are probably the result of local climatic conditions that could affect the two species differently [42]. In fact, despite that *P. perfiliewi* and *P. perniciosus* being usually sympatric species, *P.*
*perniciosus* prefers more arid regions than *P. perfiliewi* [33]. Thus, fluctuations in local climatic factors resulting in an increase in the global aridity and a longer arid period such as that shown in the study area in 2017 may have allowed *P. perniciosus* to last longer at high density levels [37]. This correlation between seasonal variations in the densities with climatic factors, mainly temperature and rainfall, has been abundantly reported [8,34,40,43]. *Phlebotomus perniciosus* density was positively correlated with temperature and negatively correlated with rainfall in Spain [42].

In the present study, we applied a kDNA qPCR technique to study *Leishmania* infection in sandfly populations captured in a human dwelling located in Siliana North district, with a focus of human VL with low incidence. No infection was found during the first year of collection, and the qPCR positivity in sandflies was very low (0.7%) during the second year of capture, with *Leishmania* infections being found only in four *P. perniciosus* specimens caught at the end of the sandfly activity season. This corroborates reports of other authors in Algeria [44] and highlights the importance of multi-seasonal entomological survey and utility of the qPCR technique in the screening of *Leishmania* infection in areas of VL low incidence. The ITS1 PCR was, however, positive only in one infected specimen out of four. This could probably be explained by lower sensitivity of ITS1-based PCR than the kDNA-based molecular tool [45,46], with the three negative specimens harboring a very low parasitic load of less than 20 parasites. However, even though we performed a real-time PCR using a TaqMan probe, ensuring more specificity to *Leishmania* [26], we cannot exclude that kinetoplast DNA of other kinetoplastids has been amplified, with the latter being occasionally found in sandflies, and with sandflies even being vectors of some of them [47,48]. The kDNA PCR is considered to be the most sensitive method for diagnosing leishmaniasis since there are about 10,000 minicircles per parasite. However, these reactions generally either amplify genus- or subgenus-specific conserved regions and do not allow species identification. Recent development of a barcoding-like approach based on high-throughput sequencing of kDNA amplicons and minicircle sequence comparisons may allow in the future a reliable identification of *Leishmania* species in kDNA qPCR-positive specimens [49,50].

On the other hand, infection rate in gravid females was significantly higher than in fed specimens, which highlights the importance of screening gravid females for epidemiological surveys as suggested by other authors [23]. Targeting gravid female specimens at the end of the sandfly activity season seems particularly relevant in the case of *Larroussius* species, which are gonotrophically concordant and do not feed again until the blood meal is digested and the eggs are laid, with the probability of detecting infective promastigotes increasing progressively during the second and third ovarian cycles [51].

In this northern area of low incidence of VL, *P. perniciosus*, the main vector of VL in Western Mediterranean countries, was the only infected species. No infection was detected in *P. perfiliewi*, although it was as abundant as *P. perniciosus* in the study site during the two-year period. Interestingly, *P. perniciosus* infection was detected at a very low level only during the second year of collection when climatic conditions were more favorable to proliferation and long seasonal activity duration of this *Larroussius* species. The ITS1 PCR was, however, positive only in one infected specimen out of four. Therefore, the identification of *L. infantum* was only possible on one specimen of *P. perniciosus*. However, this latter specimen had a very high parasite load, underlining the major involvement of this *Larroussius* species in the transmission of *L. infantum* in VL foci of Northern Tunisia with low incidence of the disease [21].

## Figures and Tables

**Figure 1 microorganisms-10-01012-f001:**
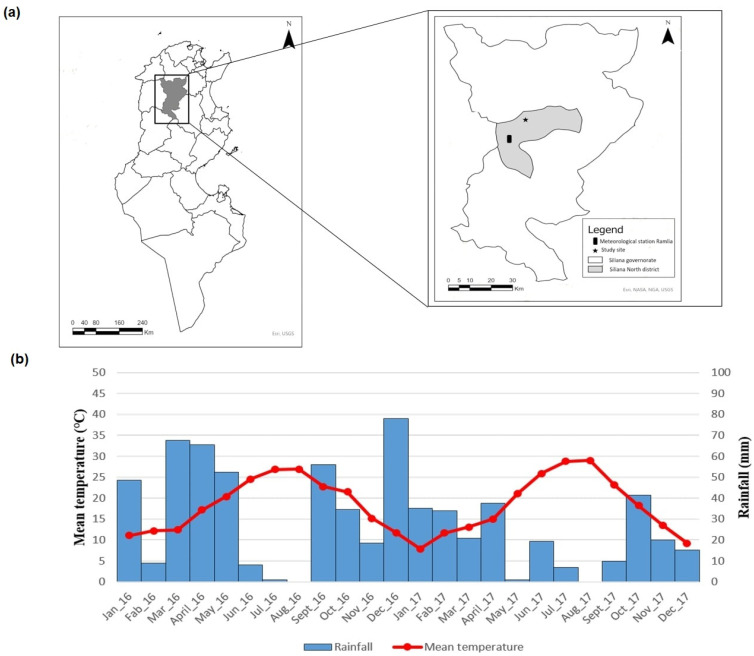
Geographical situation and ombro-thermic diagram (2016–2017) of the study area. (**a**) The map of Tunisia (**left**) shows the governorate of Siliana (shaded grey). The zoom (**right**) reports the location of Djema, the sampling site (star), and Ramlia, the meteorological station, both located in the district of Siliana-North (shaded grey). (**b**) Ombro-thermic diagram based on mean temperature and rainfall recorded in Ramlia. Arid period extended from June to August in 2016 and from May to September in 2017.

**Figure 2 microorganisms-10-01012-f002:**
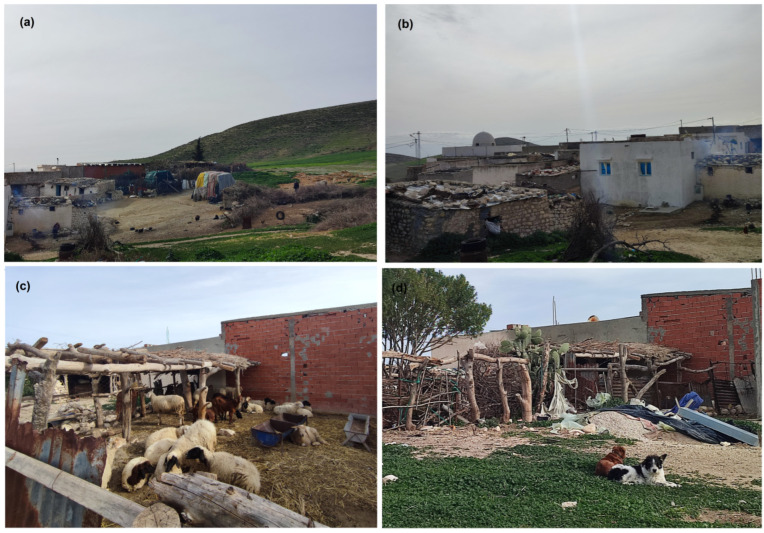
Photographs showing landscape and habitats in the sampling site of Djema. (**a**) Global view of the site: a rural zone surrounded of hills and cultivated fields. (**b**) Location of houses in relation to each other. (**c**) Animal shelter with domestic animals close to houses. (**d**) Dogs around houses and animal shelters.

**Figure 3 microorganisms-10-01012-f003:**
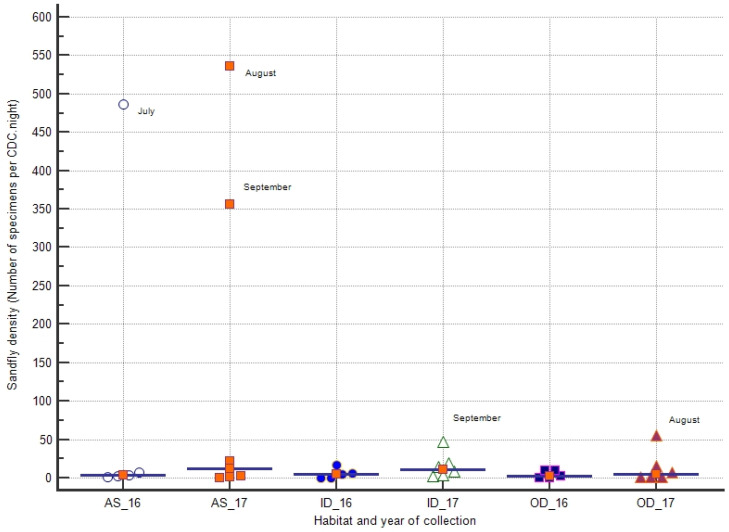
Sandfly densities according to habitat and year of capture. All monthly sandfly densities recorded by habitat and year of capture are plotted. For each group of sandfly collection, the median density is reported by a marker and a horizontal line.

**Figure 4 microorganisms-10-01012-f004:**
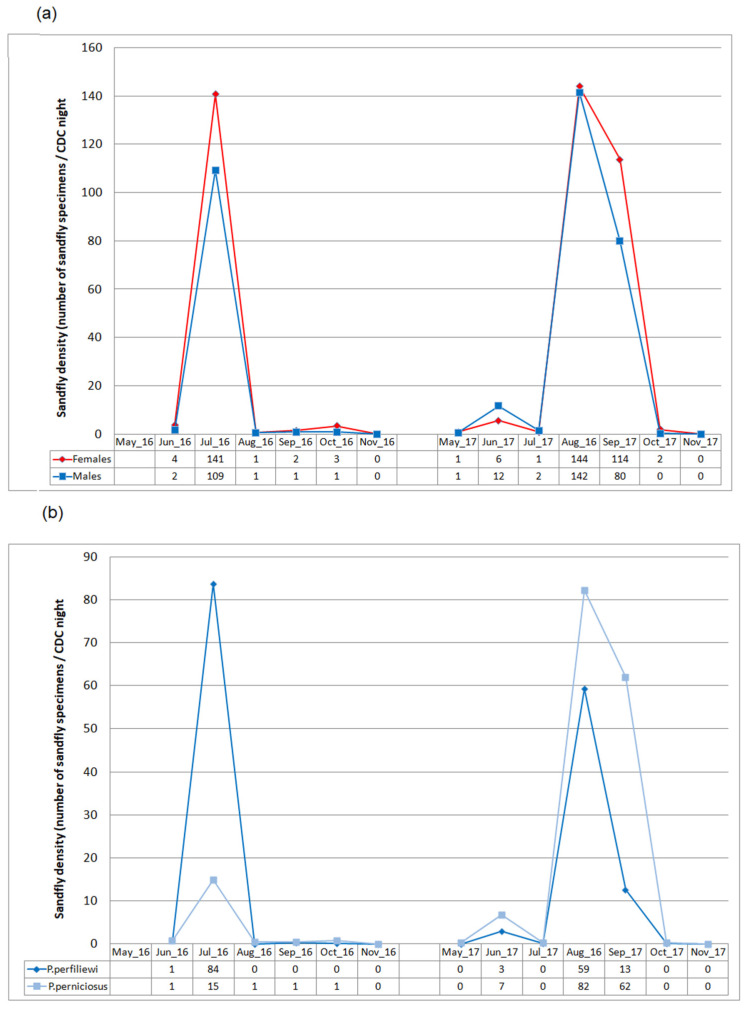
Seasonal density pattern of sandflies during the two-year period. (**a**) Density patterns of male and female sandflies. (**b**) Density pattern of *P. perniciosus* and *P. perfiliewi* male specimens.

**Table 1 microorganisms-10-01012-t001:** Number of male and female specimens according to habitat and year of capture.

Year	Habitat	Number CDC.night	Number Males	Number Females	Number Sandflies	Sex Ratio
2016	ID	12	22	37	59	0.59
	OD	12	13	27	40	0.48
	AS	24	871	1136	2007	0.77
	Total	48	906	1200	2106	0.75
2017	ID	14	111	78	189	1.42
	OD	14	60	104	164	0.58
	AS	28	1712	1954	3666	0.88
	Total	56	1883	2136	4019	0.88
Total		104	2789	3336	6125	0.84

**Table 2 microorganisms-10-01012-t002:** Physiological state of female specimens according to habitat and year of capture.

Year	Habitat	Number Females	Number Unfed (%)	Number Fed (%)		Number Gravid (%)
				Fresh BM	Digested BM	
2016	ID	37	30 (81%)	6 (16.3%)		1 (2.7%)
				1	5	
	OD	27	23 (85.2%)	4 (14.8%)		0 (0%)
				1	3	
	AS	1136	888 (78.2%)	246 (21.6%)		2 (0.2%)
				123	123	
	Total	1200	941 (78.4%)	256 (21.3%)		3 (0.3%)
				125	131	
2017	ID	78	49 (62.8%)	27 (34.6%)		2 (2.6%)
				11	16	
	OD	104	86 (82.7%)	15 (14.4%)		3 (2.9%)
				7	8	
	AS	1954	1499(76.7%)	437 (22.4%)		18 (0.9%)
				267	170	
	Total	2136	1634 (76.5%)	479 (22.4%)		23 (1.1%)
				285	194	

**Table 3 microorganisms-10-01012-t003:** Diversity of *Phlebotominae* sandfly fauna according to the sampling period and habitat.

Year	Habitat	Abundance (pi)						Diversity Indices				
		*P. perf*	*P. pern*	*P. long*	*P. pap*	*P. serg*	*S. ant*	S	H’	H’/H’max	D	D/Dmax
2016	ID	10 (0.45)	5 (0.23)	-	7 (0.32)	-	-	3	1.54	0.96	0.67	1.00
	OD	6 (0.46)	7 (0.54)	-	-	-	-	2	1.00	1.00	0.54	1.00
	AS	665 (0.76)	127 (0.15)	-	-	40 (0.05)	39 (0.04)	4	1.11	0.55	0.39	0.52
2017	ID	1 (0.01)	100 (0.90)	-	2 (0.02)	5 (0.05)	3 (0.03)	5	0.65	0.28	0.19	0.23
	OD	57 (0.95)	1 (0.02)	-	-	-	2 (0.03)	3	0.33	0.21	0.10	0.15
	AS	542 (0.32)	1115 (0.65)	3 (0.00)	-	3(0.00)	49 (0.03)	5	1.11	0.48	0.48	0.59

*P. perf*: *P. perfiliewi*; *P. pern*: *P. perniciosus*; *P. pap*: *P. papatasi*; *P. serg*: *P. sergenti*; *P. long*: *P. longicuspis*; pi: relative abundance; S: specific richness; H’: Shannon’s index; H’/H’max: equitability index; D: Simpson’s index; D/Dmax: equitability index.

**Table 4 microorganisms-10-01012-t004:** Number of infected specimens according to habitat, physiological state, and month and year of capture.

		Number of Specimens in 2016			Number of Specimens in 2017		
		Collected	DNA	+ kDNA qPCR	Collected	DNA	+ kDNA qPCR (Infection Rate)
Habitat	ID	37	32	0	78	73	1 (1.36%)
	OD	27	24	0	104	103	0
	AS	1136	255	0	1954	391	3 (0.76%)
Month of capture	May	-	-	-	7	7	0
	June	28	28	0	42	42	0
	July	1128	239	0	7	7	0
	August	5	5	0	1151	275	0
	September	12	12	0	912	223	4 (0.44%)
	October	27	27	0	17	13	0
	November	0	0	-	0	0	0
Physiological state	Unfed	941	178	0	1634	304	0
	Fed	256	130	0	479	240	2 (0.83%)
	Gravid	3	3	0	23	23	2 (8.69%)
Blood meal	Fresh	125	48	0	285	87	2 (2.29%)
	Digested	131	82	0	194	153	0


Total female specimens		1200	311	0	2136	567	4 (0.70%)

**Table 5 microorganisms-10-01012-t005:** Parasitic load, ITS1 PCR results, and *Leishmania* species identification of infected specimens.

Code	Habitat	Physiological State	Month of Capture	Estimated Number of Parasites per Sandfly Female Specimen	ITS1 PCRResult	*Leishmania* Species
Dj033E10	AS	Gravid	Sept 2017	2 parasites	Negative	
Dj033F4	AS	Gravid	Sept 2017	38,000 parasites	Positive	*L. infatum*
Dj032B1	AS	Fed	Sept 2017	4 parasites	Negative	
Dj030F1	ID	Fed	Sept 2017	18 parasites	Negative

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
