# Peer review of "Diversity, Abundance and Leishmania infantum Infection Rate of Phlebotomine Sandflies in an Area with Low Incidence of Visceral Leishmaniasis in Northern Tunisia"

_microorganisms, 2022, doi:10.3390/microorganisms10051012_

Round 1

Reviewer 1 Report

I appreciate the effort the authors have put into the first draft of the manuscript; they have taken all my comments into account. I recommend it for publication in Microorganisms. 

A few minor typos are noted below:

Some species and genus names are not italicized (lines 137, 148, 165, 228-233, chapters 3.4, 3.5).

Line 27 visceral

Line 99 the higher....the lower

Lines 375, 378: indoor, line 377 outdoor

Author Response

Some species and genus names are not italicized (lines 137, 148, 165, 228-233, chapters 3.4, 3.5).

Response: All corrections were done and highlighted in yellow. However, we noticed that some words in italics in the original document were no longer in italics in the uploaded document. We have corrected the text but there may be a formatting problem with the template. We will report this and send a copy of the revised version by e-mail for your information.

Line 28: visceral

Line 100: the higher....the lower

Response: All corrections were done and highlighted in yellow.

Lines 375, 378: indoor, line 377 outdoor

Response: This was corrected. However, as requested by the academic editor, abbreviations (ID and OD) were used consistently throughout the text

Reviewer 2 Report

The authors modified the manuscript; however, the result is basically same as previous one and still like a previous publication. Again, I could not find any novel findings to publish in this journal.

Other comments:

The authors identified only one of four parasite species, and discussed it was because of lower sensitivity of ITS PCR. ITS PCR is sensitive enough to detect parasites with sand flies; probably 1-10 parasites can be detected. As the authors mentioned, kDNA PCR is more sensitive; however, such sensitivity is not necessary in this case. In addition, you need to be careful about its specificity. Parasites from the three of four positive sand flies may not be Leishmania, but other related parasites such as Trypanosoma spp. if they cannot be really amplified by ITS PCR

Author Response

The authors identified only one of four parasite species, and discussed it was because of lower sensitivity of ITS PCR. ITS PCR is sensitive enough to detect parasites with sand flies; probably 1-10 parasites can be detected. As the authors mentioned, kDNA PCR is more sensitive; however, such sensitivity is not necessary in this case.

Response: kDNA qPCR was used for screening infection. Screen requires a highly sensitive technique

In addition, you need to be careful about its specificity. Parasites from the three of four positive sand flies may not be Leishmania, but other related parasites such as Trypanosoma spp. if they cannot be really amplified by ITS PCR

Response: specificity of kDNA PCR was discussed in the discussion section

Reviewer 3 Report

The manuscript “Diversity, abundance and Leishmania infantum infection rate of phlebotomine sandflies in an area with low incidence of visceral leishmaniasis in Northern Tunisia” describes the diversity, abundance and L. infantum infection rate of phlebotomine sandflies in an area with low incidence of visceral leishmaniasis in northern Tunisia and to discuss the usefulness of quantitative kDNA real time PCR and molecular investigation of gravid female specimens in screening Leishmania infection and determining L. infantum vectors. The paper is well structured, and the topic is very interesting. ? The final comments are inherent in the study and the need for further investigation because there is still so much to know.

However, some should be changed:

- the bibliography does not correspond in the text, it should be reviewed and adjusted.

- in paragraph 3.2 everything should be put in italics: P. (Larroussius) perniciosus, P. (Larroussius) perfiliewi Sergentomyia (Sergentomyia) antennata, P. (Paraphlebotomus) sergenti, P. (Phlebotomus) and P. (Larroussius) longicuspis; P. perniciosus and P. perfiliewi Phlebotomus longicuspis, P. papatasi, P. sergenti and S. antennata.

- line 272 p. perniciosus and P. perfiliewi in italics

Line 315 – add space after point

Line 332 - adjust the title table 5, “speciments”.

Author Response

- the bibliography does not correspond in the text, it should be reviewed and adjusted.

- in paragraph 3.2 everything should be put in italics: P. (Larroussius) perniciosus, P. (Larroussius) perfiliewi Sergentomyia (Sergentomyia) antennata, P. (Paraphlebotomus) sergenti, P. (Phlebotomus) and P. (Larroussius) longicuspis; P. perniciosus and P. perfiliewi Phlebotomus longicuspis, P. papatasi, P. sergenti and S. antennata.

- line 272 P. perniciosus and P. perfiliewi in italics

Response: The bibliography was reviewed and adjusted. In paragraph 3.2 everything was put in italic.

However, we noticed that  (i) in the list of references, some lines have been modified generating additional numbers. We have corrected the reference list but there may be a formatting problem with the template.

(ii) some words in italics in the original document were no longer in italics in the uploaded document. We have corrected the text but there may be a formatting problem with the template.

We will report these problems and send a copy of the revised version by e-mail for your information.

Line 315 – add space after point

Line 332 - adjust the title table 5, “speciments”.

Response: All corrections were done and highlighted in yellow.

Round 2

Reviewer 2 Report

The paper is not improved.

Author Response

We have responded to all the comments from Reviewer 2.

We have no new suggestions about our latest version of the manuscript. Therefore,  we are sorry, we cannot respond in any constructive way. 

This manuscript is a resubmission of an earlier submission. The following is a list of the peer review reports and author responses from that submission.

Round 1

Reviewer 1 Report

The manuscript describes a field study focused on sand fly fauna in the locality of the low incidence of VL in Northers Tunisia. The study is well designed and the collected material is sufficient to provide high-quality epidemiological data. The most surprising fact is the considerable difference between successive trapping years in the composition of the sand fly community. I have only minor comments and suggestions:

I appreciate the highly detailed description of Materials and Methods. I only put a query if authors monitored climatic/microclimatic conditions during their trapping sessions. If these data exist it would be highly interesting to find out if the climatic conditions differed substantially between 2016 and 2017 and if some climatic factors correlate with the abundance of dominant sand fly species, as is speculated in the discussion.

According to chapter 2.2., female sand flies were visually classified as unfed, fed and gravid. Were the females dissected? If not, a small volume of digested blood, as well as eggs in the first stages of development, may not be revealed and, therefore, numbers of fed and gravid females may be underestimated. This fact should be mentioned in the second paragraph of the discussion.

Lines 53-54: I suggest changing “an increasing degree of blood digestion would result in the presence of more parasites” to “later stages of blood digestion would be coincident with the presence of more parasites”

Line 76: Pinus halepensis

Lines 106 and 123: Leishmania

Figure 2 – Font size of the descriptions should be higher

Chapters 3.1. - 3.4.: Latin names of genera and species should be italicized

Line 281:  L. infantum, the causative agent

Line 305: abundance

Lines 310 and 344: indoor

Lines 336-7: What does mean the term “phlebotomine behavior “? I suggest using just “behavior” or specify – “resting behavior”, “feeding behavior” etc.

Reviewer 2 Report

This study aimed to identify sand fly distribution and the vector species responsible for the transmission of Leishmania infantum. The study was well-designed; however, I cannot find any novel findings in this study. Very similar paper and results from next district was published by the same group (PLOS One, 2017). I recommend publishing this paper in a more local journal.